# A Review of Therapies for *Clostridioides difficile* Infection

**DOI:** 10.3390/antibiotics14010017

**Published:** 2024-12-31

**Authors:** Faiza Morado, Neha Nanda

**Affiliations:** 1Department of Pharmacy, Keck Medical Center, University of Southern California, Los Angeles, CA 90033, USA; faiza.morado@med.usc.edu; 2Division of Infectious Diseases, Department of Medicine, Keck School of Medicine, University of Southern California, Los Angeles, CA 90033, USA

**Keywords:** *C. difficile* infection, recurrent *C. difficile* infection, metronidazole, vacomycin, fidaxomicin, fecal microbiota transplantation

## Abstract

*Clostridioides difficile* is an urgent public health threat that affects approximately half a million patients annually in the United States. Despite concerted efforts aimed at the prevention of *Clostridioides difficile* infection (CDI), it remains a leading cause of healthcare-associated infections. CDI is associated with significant clinical, social, and economic burdens. Therefore, it is imperative to provide optimal and timely therapy for CDI. We conducted a systematic literature review and offer treatment recommendations based on available evidence for the treatment and prevention of CDI.

## 1. Introduction

*Clostridioides difficile* infection (CDI) is caused by a spore-forming, toxin-producing, anerobic, Gram-positive Bacillus, originally named *Bacillus difficile* [1]. Since it was first discovered in 1935, this bacteria has been renamed *Clostridioides difficile* (*C. difficile*) and has emerged as one of the leading causes of nosocomial infections. The global CDI incidence rate is estimated to be 1.1 to 631.8 per 100,000 population per year [2]. While global healthcare expenditures associated with CDI are not known, attributable costs are likely significant as associated costs in the US alone are estimated to be 5.4 to 6.3 billion per year [2]. Despite enormous efforts focused on preventive strategies, *C. difficile* remains a significant healthcare burden worldwide.

Recurrent *Clostridioides difficile* infection (rCDI) occurs in 10% to 25% of individuals after an initial episode and up to 65% in individuals with more than one rCDI episode [3,4,5]. Recurrent CDI is more difficult to treat and is associated with significant morbidity and mortality. In the endemic setting, mortality rates range between 5 and 10%, which increase to 30–40% in the setting of fulminant disease [6,7,8]. While risk factors associated with recurrence are similar to those for a primary episode, prior studies have shown that worse outcomes are associated with each subsequent episode [9]. The leading cause of recurrent CDI is gut dysbiosis, which can take several forms such as reduced bacterial diversity, the loss of beneficial microbes, and pathobiont expansion [10]. A healthy microbiota can inhibit *C. difficile* from replicating and expanding in the colon, but a disruption of this healthy and diverse microbiome allows for increased host susceptibility and creates an ideal environment for CDI to occur [11,12].

Given the pronounced impact of CDI on morbidity and mortality, it is of the utmost importance to offer optimal and timely CDI treatment. Therefore, the purpose of this review article is to provide a summary of evidence-based strategies for the treatment and prevention of CDI, with a primary focus on new, novel microbiota-based therapies.

## 2. Materials and Methods

A literature search of PUBMED was conducted using the MeSH terms *Clostridioides difficile* and the treatment management or prophylaxis of *Clostridioides difficile* infection (CDI) between January 2019 and September 2024. The search included specific therapeutic options such as fidaxomicin, vancomycin, metronidazole, bezlotoxumab, fecal microbiota, live biotherapeutics, tigecycline, and rifamycin. Studies published in non-English languages, involving animals or children, pre-prints, and case reports were excluded. Bibliographies of included studies were further reviewed to identify other relevant studies.

## 3. Results

A total of 668 articles were identified following the initial PubMed search. Articles were reviewed for relevance. Following screening, 495 articles were excluded, with indications described in Figure 1. The remaining 173 articles were reviewed in depth.

### 3.1. First-Line Antimicrobials for the Treatment of CDIi in Adults

#### 3.1.1. Metronidazole

There are three widely accepted antimicrobial options for the treatment of CDI: metronidazole, vancomycin, and fidaxomicin. Although it is not an FDA-approved indication, historically, metronidazole has long been considered a first-line therapeutic option based on the 90–95% success rate documented in early clinical trials [13,14,15]. However, surveillance studies have since described an increasing trend in metronidazole minimum inhibitory concentration (MIC) values for common *C. difficile* strains, which may contribute to the increasing number of metronidazole-associated treatment failures observed over the past decade [16,17].

Based on recent evidence, metronidazole does not perform as favorably as once documented in early trials. In a large prospective, randomized, double-blind, placebo-controlled trial that compared vancomycin to metronidazole treatment in 150 patients stratified by disease severity, metronidazole and vancomycin resulted in a clinical cure in 90% and 98% of patients with mild disease (*p* = 0.36) and 76% and 97% of patients with severe disease (*p* = 0.02), respectively [18]. While vancomycin and metronidazole were equally effective in mild presentations of CDI, vancomycin was the superior choice for the treatment of severe CDI. As a result, the Infectious diseases Society of America (IDSA) and Society for Healthcare Epidemiology of America (SHEA) updated CDI treatment guidelines in 2010 to recommend against the use of metronidazole specifically for patients with severe disease due to lower clinical cure rates in this landmark trial [19].

Inferior clinical cure rates associated with metronidazole were later corroborated in two large phase 3 clinical trials. Johnson et al. first demonstrated that metronidazole had a 10% lower probability of achieving a clinical cure compared to patients treated with vancomycin (72.7% vs. 81.1%, respectively; *p* = 0.02) [20]. In addition, clinical success rates in patients with severe disease were higher in those treated with vancomycin, but the difference was not statistically significant (*p* = 0.059) [20]. Stevens et al. later demonstrated that metronidazole was inferior to vancomycin by showcasing that vancomycin use was associated with a significantly lower 30-day mortality than metronidazole (RR 0.86, 95% CI 0.74–0.98) in any severity cohort [21]. This favorable response was more evident in patients with severe CDI, in which vancomycin significantly reduced the risk of all-cause 30-day mortality by 20% (RR 0.79, 95% CI 0.65–0.97) [21].

Based on this evidence, the IDSA/SHEA CDI treatment guidelines published in 2022 recommended against metronidazole for all adult patients with CDI and instead recommend vancomycin or fidaxomicin [22]. Metronidazole should only be considered in mild cases when the other two agents are unavailable. Similarly, the American College of Gastroenterology (ACG) cites that oral metronidazole can be considered for the initial treatment of non-severe CDI in low-risk patients.

#### 3.1.2. Vancomycin and Fidaxomicin

The current cornerstone for the management of CDI involves treatment with one of the two FDA-approved medications for CDI: vancomycin or fidaxomicin. While metronidazole has historically been the drug of choice for CDI, it has been replaced by oral vancomycin for the reasons previously mentioned. The mechanistic hypotheses to explain vancomycin’s improved performance compared to metronidazole include the following: (1) nearly all strains of *C. difficile* maintain high susceptibility to vancomycin in vitro and (2) vancomycin achieves high fecal concentrations with mean fecal concentrations to MIC_90_ ratios of approximately 1000:1 using an observed MIC_90_ of 2 mcg/mL [23,24,25,26]. In contrast, metronidazole in vitro susceptibility is less predictable. In addition, it is efficiently absorbed, and therefore, the majority of the drug is delivered from the bloodstream to the inflamed colonic mucosa. In patients with severe disease, there may be a decrease in blood flow to the colon, thus decreasing colonic delivery and drug concentrations of metronidazole [18,27].

Then, in 2011, fidaxomicin received FDA approval for the treatment of CDI and was soon included in national guideline recommendations thereafter. The treatment of CDI is based on the severity of the current episode and number of prior episodes. For the treatment of the first episode of non-fulminant CDI, treatment with oral vancomycin or oral fidaxomicin have demonstrated similar clinical cure rates. Two pivotal double-blind, randomized clinical trials (RCTs) reported cure rates of fidaxomicin ranging from 87% to 92%, which were similar to those reported for patients treated with vancomycin [28,29]. While fidaxomicin and vancomycin have similar efficacy with regard to the clinical cure or resolution of acute diarrheal disease, many experts prefer fidaxomicin over vancomycin as it has been associated with the improved sustained resolution of disease, a shorter time to the resolution of diarrhea, and most importantly, a lower recurrence rate. Fidaxomicin has demonstrated a relative risk reduction in recurrence of approximately 40–50% in clinical trials, which included patients with first or second episodes of CDI [5,28,29,30,31].

Certain pharmacokinetic, pharmacodynamic and microbiological characteristics may explain the favorable results for fidaxomicin with respect to CDI recurrence. Fidaxomicin is the most potent anti-*C. difficile* agent, with an MIC range of 0.004–1 mcg/mL [26]. Additionally, fidaxomicin can achieve high fecal concentrations several magnitudes above reported MICs and with fecal concentration to MIC_90_ ratios of 5000:1 based on an MIC_90_ of 0.25 mcg/mL [32]. Furthermore, fidaxomicin has been shown to exhibit a prolonged post-antibiotic effect and has a narrower antimicrobial spectrum to mitigate disruption to the normal gut microbiome [28]. While clinical cure rates of fidaxomicin and vancomycin are similar, IDSA/SHEA CDI treatment guidelines preferentially recommend fidaxomicin for the treatment of the first episode of non-fulminant CDI. Comparatively, the ACG recommends either vancomycin or fidaxomicin based on comparable efficacy data and the lower cost associated with vancomycin [33]. A comparison of IDSA/SHEA and ACG treatment recommendations for CDI is in Table 1.

Despite successful initial treatment with fidaxomicin or vancomycin, recurrent CDI can occur in 10–25% of patients after the initial episode and up to 65% in patients after the first recurrence [3,4,5,34]. Two randomized clinical trials evaluated the clinical efficacy of fidaxomicin 200 mg twice daily and vancomycin 125 mg four times daily for 10 days stratified by the number of CDI episodes. While both trials included patients with either no prior episode or one episode of CDI in the past 3 months, approximately 85% of patients were enrolled with primary CDI versus approximately 15% with rCDI [28,35]. Given the small number of patients in the rCDI subgroup, both studies were underpowered to detect a difference in recurrence rates amongst patients with a history of one prior CDI. A significant reduction in rCDI with fidaxomicin was only found in patients with no prior CDI episode (i.e., primary CDI). In patients with one prior CDI episode, only a non-significant trend towards lower recurrence rates was associated with fidaxomicin [28,35]. The results of these studies suggest that fidaxomicin is optimally beneficial for recurrence prevention in patients with no prior CDI but can also be considered for patients with one prior episode given the trends towards lower recurrence. The data from RCTs for fidaxomicin in patients with multiple episodes of recurrence are limited.

Guery et al. compared an extended-pulsed fidaxomicin regimen (200 mg twice daily on days 1–5, then once daily on alternate days on days 7 to 25) to vancomycin 125 mg four times daily on days 1–10. This study included patients with up to two prior CDI episodes; however, only 5.6% of patients in this subgroup were ultimately enrolled. The primary endpoint was a sustained clinical cure 30 days after the end of treatment. Extended-pulsed fidaxomicin was superior to standard-dose vancomycin for a sustained clinical cure (70% (124/177) extended-pulsed fidaxomicin vs. 59% (106/179) vancomycin (OR 1.62 95% CI 1.04–2.54, *p* = 0.030)) [5]. The recurrence of CDI at day 90 was also lower in the fidaxomicin group (6% (11/177) vs. 19% (34/176), OR 0.29 (95CI 0.14–0.60; *p* = 0.00073) [5]. While the number of patients with two previous CDI episodes was limited, this study does demonstrate the potential benefit of a fidaxomicin extended-pulsed regimen in this subpopulation.

In addition to fidaxomicin, other guidelines recommend options for the treatment of second or subsequent recurrence that include oral vancomycin therapy using a tapered and pulsed regimen (e.g., 125 mg four times daily for 10–14 days, two times per day for a week, once per day for a week, and then every 2 or 3 days for 2 to 8 weeks). This dosing regimen is based on limited evidence and largely supported by one RCT. In this trial, most patients had a history of four to five episodes (ranging from two to nine) prior to randomization. Patients received a 6-week oral vancomycin taper or 14 days of oral vancomycin followed by fecal transplantation enema. There was no difference in rCDI in the cohort receiving the fecal transplant enema or tapered oral vancomycin course [56.2% (9/16) fecal transplant vs. 41.7% (5/12) vancomycin taper] [36].

Based on the available evidence, antimicrobial treatment recommendations for recurrent CDI include standard-dose fidaxomicin, extended-pulsed fidaxomicin, standard-dose vancomycin, or vancomycin tapered and pulsed (see Table 1). Fidaxomicin can be considered as the preferred therapeutic option for an initial CDI episode or first recurrence. For second recurrences, fidaxomicin extended-pulsed or vancomycin tapered and pulsed regimens can be considered. Given the limited data for fidaxomicin in multiple recurrences, one should consider vancomycin pulsed and tapered regimens for three or more recurrences in addition to the adjunctive or microbiota-based therapies discussed below.

For rCDI, the choice between therapeutic options often depends on the antibiotic used for the first episode [37]. For example, if the first episode of CDI was treated with a standard course of oral vancomycin, the first rCDI would be treated with standard fidaxomicin or a vancomycin tapered and pulsed regimen. In addition, risk factors for recurrence such as an age greater than 65 years, concomitant systemic antibiotics for non-CDI indication, comorbidities, and severe infection should be taken into account. Guery et al. demonstrated that an extended pulsed regimen of fidaxomicin had lower recurrence rates compared to vancomycin and was particularly advantageous for patients at high risk of recurrence [5]. Patients were included in this study if they were aged 60 years or older, and many presented with concomitant risk factors for rCDI such as severe CDI, a history of one or two prior CDI episodes, concomitant systemic antibiotics for a condition other than CDI, and documented cancer in their medical history [5]. If patient compliance allows, an extended pulsed regimen of fidaxomicin could be preferentially considered for patients at the highest risk for recurrence.

For fulminant CDI, patients are treated with a higher dose of oral vancomycin plus intravenous metronidazole and vancomycin rectal enema if an ileus is present [22,33]. Fidaxomicin clinical trials have excluded patients with life-threatening or fulminant CDI. In the absence of supportive data, fidaxomicin is not recommended for the treatment of fulminant CDI. While studies comparing standard-dose vancomycin (i.e., 125 mg four times daily) to high-dose vancomycin (i.e., 500 mg four times daily) suggested that there was no difference in clinical cure rates, these studies excluded patients who presented with fulminant CDI [25,38,39]. Rather, this recommendation is based on prudency. An ileus reduces gastrointestinal motility and delays the GI transit of oral medications. Additionally, profuse diarrhea with a higher stool frequency can reduce the contact time of vancomycin in the colon [24]. Therefore, patients with an ileus in the setting of profuse diarrhea may benefit from high doses of oral vancomycin. Similarly, there are limited data to support the use of rectal vancomycin for patients with an ileus. Given the severity and urgency of fulminant CDI, rectal vancomycin and intravenous metronidazole are recommended to ensure therapeutic concentrations within an inflamed colon [22].

### 3.2. Alternative Antimicrobial Therapies

#### 3.2.1. Rifaximin

Rifaximin, a non-absorbable rifamycin, is currently indicated for the treatment of traveler’s diarrhea and reducing the recurrence of hepatic encephalopathy. Since it displays potent activity against *C. difficile* and achieves high colonic concentrations because of its poor absorption, rifaximin has been explored as a treatment option for other GI infections like CDI [40]. Rifaximin has been most studied as an adjunctive or “follow-on” agent after vancomycin treatment in patients with rCDI.

The IDSA/SHEA guidelines recommend a standard course of oral vancomycin followed by rifaximin in patients with >1 recurrence [22]. This is largely based on a small RCT in which patients received rifaximin 400 mg three times daily for 20 days after standard CDI therapy with vancomycin or metronidazole. Recurrent CDI occurred in 5 of 33 (15%) patients given rifaximin and 11 of 35 (31%) patients given a placebo (*p* = 0.11), which was not statistically significant due to the small sample size [41]. While there was a numerical reduction in rCDI, the difference was not significant. While rifaximin may be promising, it has not been definitively proven as effective. Lastly, the results of these trials cannot be extrapolated to patients treated with fidaxomicin. As the standard-of-care treatment changes from metronidazole and vancomycin to fidaxomicin (an anti-CDI therapy associated with lower recurrence rates), the benefit of rifaximin may be even less impressive.

One concern associated with the use of rifaximin is the potential for resistance. *C. difficile* isolates with elevated MICs >256, and isolates developing high MICs during rifaximin therapy have been reported, with resistance rates ranging from approximately 30 to 50% [40]. In the absence of a definitive demonstration of benefit and concerns for resistance, the ACG suggests that further randomized trials, including cost–benefit analyses, are needed. Therefore, the ACG does not currently recommend its routine use [33].

Another consideration for rifaximin is its effect on the gut microbiome. In addition to potent activity against *C. difficile*, it is more potent than fidaxomicin and vancomycin against Gram-negative anerobic bacteria such as *Bacteroides* spp. As a result, gut dysbiosis is further exacerbated, thus increasing the risk of recurrence [42]. Given the paucity of data, concerns for resistance, and propensity to disrupt the gut flora, it may be more beneficial to explore alternative options for patients with >1 recurrence such as monoclonal antibodies or microbiota-based therapies in addition to standard-of-care antibiotics.

#### 3.2.2. Tigecycline

Tigecycline, a tetracycline derivative, is a broad-spectrum antibiotic that is active against various Gram-positive and Gram-negative aerobic and anerobic bacteria including *C. difficile*, *Bacteroides fragilis*, *Prevotella* spp., and multi-drug resistant bacteria [43]. As a result, it can have a significant impact on the gut microbiome and subsequently increase the risk of rCDI.

Both the IDSA/SHEA and ACG guidelines describe tigecycline as an alternative agent with its suggested efficacy in the treatment of *C. difficile*, but they do not recommend its routine use. There are no RCTs to date that have evaluated the safety or efficacy of tigecycline for the treatment of CDI. The current evidence to support the use of tigecycline is limited to retrospective and observational studies in severe or fulminant CDI [44,45,46,47]. A meta-analysis of four studies that included patients with severe CDI and treated with tigecycline as monotherapy (45/186, 24.12%) or in combination with other antibiotics (141/186, 75.88%) reported a pooled clinical cure rate of 79% (95% CI 73–84.5%) [43]. While tigecycline may be a promising agent, the authors of the meta-analysis acknowledged the lack of RCTs and heterogeneity of the included studies and suggested that further studies are needed to elucidate the role of tigecycline in the treatment of CDI. A phase 2 clinical trial was started in 2011 to specifically address this question but was discontinued due to its slow enrollment rate.

While there is insufficient evidence to support the universal use of tigecycline, it may be considered as part of a salvage regimen in patients with severe or fulminant CDI based on a retrospective study that demonstrated higher clinical cure rates for patients treated with tigecycline compared to standard antibiotics for severe or fulminant CDI (75.6% vs. 53.3%, respectively; *p* = 0.02). In this study, the majority of patients (84.6%) received tigecycline after failure with vancomycin plus metronidazole, thus suggesting its role as salvage therapy [47]. In the absence of RCTs, risk may outweigh the potential benefit of tigecycline in the setting of non-salvage therapy.

### 3.3. Adjunctive Treatment

#### Bezlotoxumab

Bezlotoxumab is a monoclonal antibody that binds to *C. difficile* toxin B and is administered as a single dose of 10 mg/kg based on actual body weight infused over 1 h in conjunction with standard-of-care antibiotics [48]. It is not an antibiotic and therefore is not indicated for the treatment of CDI and should not be used as monotherapy. Rather, it received approval in 2016 as an adjunctive agent to standard-of-care-antibiotics to reduce rCDI in adults and pediatric patients 1 year of age and older.

Bezlotoxumab was originally studied in two double-blind, randomized, placebo-controlled phase 3 trials, MODIFY I/II, in which 2559 patients were treated with a placebo or bezlotoxumab. The administration of bezlotoxumab occurred anywhere between 1 day prior to the initiation of standard-of-care antibiotics and up to day 14 of antibiotic therapy. The distribution of standard-of-care antibiotics included 46.7% metronidazole, 47.7% vancomycin and 3.6% fidaxomicin [49]. Nearly half of the study participants had one or more prior CDI episodes. Recurrence within 12 weeks of bezlotoxumab infusion occurred in 16.5% (129/781) of patients receiving bezlotoxumab versus 26.6% (206/773) of patients in the placebo group (*p* < 0.0001). This corresponds to a number needed to treat (NNT) of 10 to prevent a single recurrence [49]. Notably, the choice of antibiotic therapy for CDI did not influence bezlotoxumab efficacy.

A post hoc analysis later investigated bezlotoxumab efficacy in patients with the following risk factors for recurrence: age > 65 years, a history of CDI, immunocompromised, severe CDI, and ribotype 027/078/244. In this post hoc analysis, 1554 patients were included, and the majority (75.6%) had at least one pre-specified risk factor for rCDI. There was no difference in initial cure rates in the bezlotoxumab and placebo group for patients with at least one risk factor and in patients with no risk factors [50]. However, bezlotoxumab reduced the rates of rCDI amongst patients with any of the five pre-specified risk factors. The absolute reduction in rCDI associated with bezlotoxumab increased with the number of risk factors present: −14.2%, −14.2% and −24.8% for those with 1, 2 and 3 or more risk factors, respectively (95% CI excluded 0 for all comparisons) [50]. The results from this study suggest that bezlotoxumab is particularly effective in patients who are at a high risk of the recrudescence of disease, which include those who are 65 years or older, have a history of CDI, or are infected with hypervirulent ribotypes 027/078/244.

The second post hoc analysis evaluated the timing of bezlotoxumab administration with respect to antibiotic treatment initiation for CDI. In 1554 patients included in this analysis, 649 (41.8%), 469 (30.1%), and 436 (28.1%) received an infusion at 0–2, 3–4, and >5 days after the initiation of anti-CDI therapy. The rate of clinical cure and the time to the resolution of diarrhea were similar in all groups irrespective of the timing of administration (range 77.8 to 81.4% bezlotoxumab vs. 77.8 to 81.7% placebo) [51]. The results of this study suggest bezlotoxumab can be administered at any point during antibiotic therapy for CDI.

Based on the available evidence, bezlotoxumab can be considered for patients with one or more of the following risk factors for rCDI: age > 65, a history of CDI, immunocompromised, severe CDI, prior CDI, or the presence of a hypervirulent C. difficile strain. This coincides with current CDI guideline recommendations. ACG specifically recommends the addition of bezlotoxumab for patients who are at a high risk of recurrence. IDSA/SHEA recommends the addition of bezlotoxumab in patients with a CDI episode within the past 6 months for first or multiple recurrences and for patients at high risk for CDI recurrence for primary CDI.

When considering the addition of bezlotoxumab, safety considerations should be made. While adverse events (AEs) in clinical trials are generally mild, with symptoms of nausea, headache, fatigue, dizziness, and pyrexia occurring in similar rates between bezlotoxumab and the placebo (7.5% and 5.9%, respectively), the package insert does provide additional AEs and mortality rates for patients with congestive heart failure (CHF). In patients with a history of CHF, a serious reaction of heart failure occurred in 12.7% of bezlotoxumab-treated patients compared with 4.8% of patients in the placebo arm. In these same patients, the mortality rate was higher in the bezlotoxumab group compared to placebo (19.5% vs. 12.5%, respectively) [48]. Furthermore, patients with CHF were more likely to report increased treatment-emergent adverse events (83.9% vs. 12.5%, respectively) and serious adverse events (53.4% vs. 38%, respectively) [33]. While there is no absolute contraindication to the use of bezlotoxumab, it may be best to avoid bezlotoxumab in patients with CHF out of prudency.

### 3.4. Microbiota Restoration Therapies for the Prevention of CDI

#### 3.4.1. Fecal Microbiota Transplantation (FMT)

Fecal microbiota transplantation (FMT) was first used in the 4th century in China to treat patients with severe diarrhea and food poisoning [52]. It was first reported in the medical literature in 1958 when it was successfully used for the treatment of pseudomembranous colitis. The practice of FMT gained traction following the landmark open-label randomized trial in Netherlands in 2013. This study showed that the duodenal infusion of donor feces preceded by a short vancomycin regimen and bowel lavage (13/16, 81% cured) was superior to vancomycin alone (4/13, 31%) or vancomycin and bowel lavage (3/13, 23%) in patients with recurrent CDI [53].

Over the years, several studies have shown the benefits of FMT in many conditions, including recurrent CDI, inflammatory bowel disease, alcoholic liver disease, cirrhosis, Parkinson’s, Alzheimer’s disease, and metabolic syndrome to name a few [54]. A Cochrane systematic analysis conducted to assess the safety and efficacy of FMT in recurrent CDI in adults showed the superior efficacy of FMT over other treatment modalities. This systematic analysis included six randomized trials including 320 participants from Canada, Italy, the Netherlands and the United States. These trials excluded immunosuppressed individuals. The pooled results from the six studies showed that the use of FMT with recurrent CDI resulted in a higher rate of resolution of recurrent CDI compared to other controls, with a risk ratio of 1.92 and confidence interval (CI) 1.36 to 2.71 with statistical significance. The follow-up time period after treatment with FMT was 8–18 weeks. The number needed to treat for an additional benefit outcome was only three. The amount of stool, type of donor, route of administration, and number of administrations varied across the studies. The commonly reported mild adverse events in the FMT group were abdominal pain, bloating and diarrhea. The authors were not able to draw a conclusion about safety as the number of serious adverse events was small [55]. Accordingly, the IDSA guidelines (2021) recommend using FMT after at least two recurrences if the appropriate antibiotics have failed [22].

At the patient-level, the outcome of FMT is influenced by donor- and recipient-related factors, such as the microbiota richness, underlying disease state and genetic make-up of an individual. In addition, FMT protocols regarding recipient preparation for the administration of the product also influence the final outcome of FMT. It is a well-accepted practice to administer oral vancomycin before FMT for recurrent CDI for priming the recipient’s gut for FMT. However, there is a lack of standardization for other factors like the bowel-cleansing regimen, number of fecal infusions, amount of infused feces and route of delivery. Based on limited evidence, for recurrent CDI, higher cure rates were achieved with repeated FMT compared to a single infusion [56].

The mechanism of action of FMT in recurrent CDI remains unclear. Potential mechanisms that have been proposed include the restoration of microbial ecology and favorable changes in microbial-derived metabolites. The restoration of microbial ecology implies an increase in microbial diversity, which results in an increase in the colonization resistance against *C. difficile*. Microbial metabolites that are known to play an important role in CDI pathology are bile acids and short-chain fatty acids (SCFAs). Primary bile acids (PBAs) promote *C. difficile* germination and secondary bile acids inhibit spore germination. In individuals with recurrent CDI, there is an excess of PBA and diminished secondary bile acids. Following FMT, there is a restoration of bile acid composition that resembles that of healthy donors. Other metabolites, SCFAs, are known to be protective against *C. difficile*, i.e., the higher their level, the more protection they offer against CDI, and their levels are restored following FMT. Other less studied mechanisms of FMT are the immune-mediated mechanism, epigenetic-related mechanism, and impact on the gut–liver–brain axis [54,57]

Given the limited understanding of the mechanism of action of FMT, there is variability in regulating its use globally. For instance, in the USA and Canada, FMT is an investigational drug that can be used to treat recurrent CDI and in the context of clinical trials for other diseases. However, in certain countries like Italy, the Netherlands and Belgium, it is considered a tissue transplant product, and it is a regulated medicinal product in the United Kingdom [56]. Despite this variability, the FDA in the USA has published extensive guidelines regarding the screening of FMT donors and requires the submission of details of the specific chemistry, manufacturing and control of the product before administration [58]. With these regulations in place, feces for FMT can be obtained from stool banks, the largest being OpenBiome, which collects, screens and stores stool from health donors.

FMT is an effective and relatively safe option for individuals with recurrent CDI and has an evolving role in treating chronic conditions like inflammatory bowel diseases.

#### 3.4.2. Novel Live Biotherapeutic Products

Although FMT is generally safe, it is associated with some risk. The FDA has issued a safety alert regarding the potential for the transmission of serious or life-threatening infections with pathogenic and multi-drug-resistant organisms with the use of FMT [59,60]. This underscores the need for a rigorous and standardized approach to donor qualification, pathogen screening, and the application of quality control measures to reduce the risk of transmission.

Based on a clear need for the standardization of manufacturing and administration processes, there has been a new development around standardized microbiota restoration therapies in capsule and enema form. In comparison to FMT methodologies that involve the infusion of the whole stool, newer live biotherapeutics products (LBPs) provide a smaller, more refined consortium of key bacteria with a standardized and consistent composition, concentration, and screening process for infectious organisms [61]. The recent US Food and Drug Administration (FDA) approval of LBPs has expanded patient access to microbiota restoration therapies beyond FMT. Similar to FMT, the aim of LBPs is to reconstitute the microbiome and achieve engraftment, in which healthy bacteria replicate in the recipient colon to create an inhibitory environment for *C. difficile* growth. Real-world experience with new LBPs for the prevention of rCDI will help inform its place in therapy over the next few years. The following section will review the new biotherapeutics for the prevention of rCDI: fecal microbiota, live-jslm; fecal microbiota spores, live-brpk; and VE303.

#### 3.4.3. Fecal Microbiota, Live-Jslm (Rebyota^TM^)

First, in the new class of LBPs for CDI is rectal live-jslm fecal microbiota suspension, previously known as RBX260 in clinical trials. Rectal live-jslm received FDA approval for the prevention of rCDI following antibiotic treatment for recurrent CDI. The product is administered rectally as a single 150 mL dose to be administered by a licensed healthcare provider 24 to 72 h after the last dose of antibiotics [62]. No bowel preparation is required prior to administration. The use of bowel preparation has not been studied in clinical trials to help simplify the administration process and improve patient experience [63]. Rectal live-jslm is a fecal microbiota suspension derived from healthy human stool samples that undergoes a panel screen for transmissible pathogens. In contrast to FMT in which whole stool samples are administered, fecal microbiota (FM), live-jslm provides a smaller, more refined consortium of key bacteria with a consistent make-up. Each 150 mL dose contains between 1 × 10^8^ and 5 × 10^10^ colony-forming units (CFUs) of fecal microorganisms per mL, including more than 1 × 10^5^ CFU/mL of Bacteroides, which is suspended with a solution of polyethylene glycol (PEG) 3350 and 0.9% sodium chloride in a predefined ratio [59,61]. FM, live-jslm is stored either in an ultracold freezer (−60 °C to −90 °C) or in a refrigerator (2 °C to 8 °C) if administered within 5 days [62]. Prior to its administration, healthcare personnel should ensure the suspension is warmed to room temperature. For the minimization of cramping and expulsion, patients should be instructed to remain in a left-sided prone or knee–chest position for up to 15 min post administration. A summary of drug administration information is found in Table 2. 

A safety concern with FMT is the lack of a standardized approach to donor qualification and pathogen screening processes. FM, live-jslm, however, was developed under the FDA’s Investigational New Drug program with the intent to meet the stringent requirements for approval as an FDA-regulated drug product to reduce rCDI. As a result, stool donors for FM, live-jslm undergo a rigorous screening process with routine blood and stool testing to identify pathogens such as HIV, hepatitis A/B/C, syphilis, SARS-CoV-2, enteropathogenic *Escherichia coli*, Shiga toxin-producing *E. coli*, norovirus, rotavirus, adenovirus, vancomycin-resistant enterococci, methicillin-resistant *Staphylococcus aureus*, and other antibiotic-resistant bacterial strains [64].

FM, live-jslm was approved based on data from the PUNCH CD trial series. First, in the series was PUNCH CD, which was first published in March 2016. The objective of this study was to evaluate the safety and durability of FM, live-jslm in patients with at least two rCDI episodes or at least two severe episodes requiring hospitalization. Of the 188 reported AEs, the most cited were mild to moderate in severity, primarily GI, and all self-limited: diarrhea 24%, flatulence 14%, abdominal pain and cramping 13%, and constipation 13% [65]. Twenty serious adverse events were reported but were found to be unrelated to the study drug. The resolution of CDI-associated diarrhea at 8 weeks for patients receiving either one or two doses occurred in 27/31 (87.1%) study participants [65]. Of the 14 patients who received a second dose and were available for follow-up, 78.6% (11/14) were considered treatment successes. Therefore, in all 31 patients included in this study, 87.1% (27/31) experienced a resolution of CDI-associated diarrhea. The results of PUNCH CD illustrated the short-term safety of FM, live-jslm and demonstrated similar efficacy to those reported for FMT.

In October 2022, the results from PUNCH CD2, a phase 2b randomized, placebo-controlled trial, were published. Eligible patients included those with at least three episodes of rCDI and who had received at least two courses of CDI-directed antibiotic therapy. Following a 24 to 48 h wash-out period after antibiotic therapy, the patients received a single dose of FM, live-jslm and were eligible to receive a second dose if rCDI was suspected less than 8 weeks after receiving the first dose. The study participants were randomized into one of the following treatment groups: two doses of FM, live-jslm, two doses of a placebo, or one dose of FM, live-jslm followed by one dose of the placebo. Non-responders were eligible to receive up to two doses of FM, live-jslm in the open-label part of the study. In the final intention-to-treat (ITT) analysis, a clinical cure at 8 weeks occurred in 56.8% (25/44) of patients who received one dose of FM, live-jslm and 43.2% (19/44) of patients who received one dose of the placebo (*p* = 0.201) [66]. Across all analyses, two doses of FM, live-jslm were not associated with improved treatment success. The combined efficacy for all patients who received at least one dose of FM, live-jslm, which included blinded or open-label administration, was 88.8%. Treatment-related AEs were similar across all groups during the 24-month follow-up period [66]. Although the clinical trial did not meet its pre-defined primary endpoint for treatment success observed at 8 weeks after two doses were received, a clinically meaningful and statistically significant difference was found between one dose compared to the placebo. As a result, a single-dose regimen was pursued in the phase 3 clinical trial, Punch CD3.

The Punch CD3 trial was a randomized, double-blind, placebo-controlled trial that compared one dose of FM, live-jslm (n = 180) to a placebo (n = 87) for treatment success at 8 weeks, defined as the absence of *C. difficile* infectious diarrhea. The study included patients who had one or more rCDI episodes. If treatment failure was noted within 8 weeks of study treatment, the participants were able to receive an open-label treatment of FM, live-jslm. This study used a Bayesian primary analysis that combined the results from the placebo and one-dose arm of Punch CD2 with those from matching arms of the Punch CD3 trial. Based on the Bayesian analysis integrating Punch CD2 trial data, treatment success occurred in 70.6% FM, live-jslm vs. 57.5% placebo (13.1% treatment difference), with a posterior probability of superiority of 0.991, exceeding the prespecified cutoff of 0.975 [64]. Of the patients with documented success at 8 weeks, 92.1% experienced sustained clinical resolution at 6 months. In total, 65 study participants (n = 41 FM, live-jslm; n = 24 placebo) with treatment failure received a dose of open-label FM, live-jslm. Notably, 53.7% (22/41) of participants who received two doses of FM, live jslm were deemed treatment successes within 8 weeks, and of these responders, 86% (19/22) exhibited a sustained clinical response at 6 months. Overall, 83.6% (148/177) who received blinded FM, live-jslm achieved treatment success by their second dose. During the 6-month follow-up period, a higher rate of AEs was reported in patients who received FM, live-jslm compared to the placebo (100/180, 55.6% vs. 39/87, 44.8%, respectively) [64]. However, the difference was largely driven by mild adverse events, which mainly occurred during the first 2 weeks after treatment. Punch CD3 demonstrated the superiority of FM, live-jslm compared to the placebo with a sustained clinical response up to 6 months, with no reported serious AEs.

Given the similarities between Punch CD2 and Punch CD3 that allowed for a Bayesian modeling approach, Feuerstadt and colleagues aimed to identify patient and treatment characteristics that may have impacted the safety and efficacy of FM, live-jslm. The goal of this study was to help inform real-world clinical decision making. FM, live-jslm significantly reduced rCDI in patients without T2DM, CKD, and CHF as well as those who received oral vancomycin courses >14 days. The most robust reductions in rCDI were observed in patients with a 3-day antibiotic wash-out period (24% (95% CI 1.3–46.5)) and participants with >four previous CDI episodes (20.8, 95% CI 3.3–38.0) [63]. While FM, live-jslm can be administered within 24 to 72 h of completing standard-of-care antibiotics based on FDA-approved labeling, an antibiotic wash-out period of at least 3 days may suggest the optimal clearance of antibiotics within the system to prevent unintended harm to microbiota-based therapies.

Across all three trials of Punch CD, Punch CD2, and Punch CD3, FM, live-jslm was demonstrated to have a positive benefit–risk profile for the prevention of rCDI. However, patients with the following conditions were excluded from the trial series: immunocompromised, prior FMT, pregnancy, other concurrent infections, gastrointestinal comorbidities (e.g., irritable bowel syndrome (IBS), inflammatory bowel disease (IBD), celiac disease), and liver cirrhosis. Although it may be necessary to exclude these populations in early trial designs, these patient groups would have the most to benefit from LBPs like FM, live-jslm. Addressing this concern, Punch CD3-OLS was recently published in May 2024. This study was a prospective, phase 3, open-label study designed to assess the 6-month AE rate of FM, live-jslm in patients previously excluded from the prior Punch CD trials. Secondarily, it evaluated treatment success at 8 weeks and sustained clinical response for up to 6 months. Study participants included individuals with GI comorbidities (i.e., ulcerative colitis, Crohn’s disease, IBD, IBS, GERD), mild to moderate immunocompromising conditions, and renal and urinary comorbidities. Overall, 793 patients were enrolled, in which approximately half were >65 years of age. The results of this study demonstrated an 8-week symptom resolution rate of 73.8% and a 91% sustained clinical response at 6 months with FM, live, jslm, which were comparable to the Punch CD3 RCT [67]. The majority (121/151, 80%) of the individuals with treatment failure at week 8 elected to receive a second dose of FM, live-jslm. Following the second dose, 55.4% of patients achieved treatment success [67]. Similar to previous trials, the AEs were mild to moderate and resolved with time. Punch CD3-OLS provided safety and efficacy data in a “real-world” population at higher risk for rCDI.

FM, live-jslm has consistently demonstrated to be safe and effective across several randomized controlled trials amongst patients with one or more recurrences of CDI, including complex patients such as those who are immunocompromised. Success rates are largely comparable to the ranges reported for FMT. However, unlike FMT, FM, live-jslm is manufactured under standardized processes and is an FDA-approved drug product. Based on its long-term safety and efficacy, it can be considered as an alternative to FMT in patients with at least one recurrence. While FDA labeling suggests it can be administered as early as 1 day after the completion of standard-of-care antibiotics, in the absence of bowel preparation, it may be prudent to wait 72 h to limit the lingering presence of antibiotics that may reduce its efficacy. While one dose of FM, live jslm was selected for phase 3 studies, a second dose may be considered for treatment failures after symptom recurrence within 8 weeks of the first dose.

#### 3.4.4. Fecal Microbiota Spores, Live-Brpk (Vowst^TM^)

Fecal microbiota spores, live (FMSL)-brpk, previously referred to as SER-109 in clinical trials, is another novel FDA-approved LBP indicated to prevent rCDI following standard-of-care antimicrobial treatment. FMSL-brpk is the first capsulated and orally administered fecal microbiota-based LBP that is composed of primarily live, purified Firmicute spores [68]. A reduction in *Firmicute* spp. and their key metabolites is one mechanism believed to facilitate CDI recurrence. A reduction in *Firmicute* spp. in the gut microbiome leads to an increase in primary BAs, promoting favorable conditions for *C. difficile* spore germination. As a result, the administration of live purified firmicute spores is thought to resist and limit the *C. difficile* lifecycle [69].

The manufacturing process for FMSL-brpk is quite rigorous and first involves a donor screening process that includes a detailed past medical history, physician examination, and comprehensive laboratory testing [68]. A healthy donor stool then undergoes testing for transmissible pathogens and undergoes processing, compliant with good manufacturing processes. During the purification process, the fecal matter undergoes treatment with high concentrations of ethanol to selectively kill non-Firmicutes spores, including pathogenic bacteria. Following ethanol treatment, the fecal matter undergoes filtration to remove solids and residual ethanol to isolate the firmicutes spores. The rigorous manufacturing process for FMSL-brpk results in an inactivation of several potential pathogens, fungi, parasites, and viruses, including SARS-CoV-2, and results in a standardized combination of Firmicutes spores, with each capsule containing 1 × 10^6^ and 3 × 10^7^ Firmicutes spore colony-forming units [68].

The product is supplied as capsules and recommended to be stored in the original packaging at room temperature (2 to 25 °C) [70]. The dosage of FMSL-brpk is four capsules orally once daily for 3 consecutive days on an empty stomach. Prior to taking the first dose, the patient is instructed to complete antibacterial treatment for rCDI for 2 to 4 days. In order to flush out any residual antibiotics within the patient’s gastrointestinal system that may impair FMSL-brpk activity, patients should drink 296 mL (10 oz) of magnesium citrate on the day before and at least 8 h prior to taking the first dose of FMSL-brpk. For patients with impaired renal function, polyethylene glycol electrolyte solution may be used as an alternative [70]. Except for small amounts of water, patients should not eat or drink for at least 8 h prior to the administration of the first dose. A summary of drug administration is found in Table 2. 

The safety and efficacy of FMSL-brpk was evaluated in the ECOSPOR trial series. Early in the trial series was ECOSPOR, a phase 2, randomized double-blind, placebo-controlled trial that included patients who had >three episodes of CDI within 9 months. Patients were randomized to receive a single dose of FMSL-bprk (n = 59) or a placebo (n = 30). rCDI up to 8 weeks after treatment, safety, engraftment and bile acid changes were analyzed. No significant difference in rCDI between FMSL-brpk and placebo was identified (44.1% vs. 53.3%, respectively; RR 1.2, 95% CI 0.8–1.9) [71]. However, in a pre-planned analysis by age stratum, FMSL-brpk significantly reduced recurrence in those aged >65 years (45.2% vs. 80%, respectively, RR 1.77; 95% CI 1.11–2.81). Notably, no benefit was shown in those aged <65 years. FMSL-brpk was generally well tolerated, with AEs occurring at similar rates in the study and placebo group. GI-based AEs were most commonly reported. Engraftment was assessed by evaluating the number of dose species in stool samples. Those receiving FMSL-brpk had more spore-forming *Firmicutes* spp. compared to the placebo throughout the 8-week follow-up (*p* < 0.001). Additionally, to measure the impact of FMSL-brpk on non-dose species, the amount of Bacteroides was assessed, and a greater abundance of Bacteroides was found in the group receiving FMSL-brpk (*p* = 0.04) [71]. To understand the relationship between engraftment and non-recurrence, the authors evaluated the relationship between engraftment and the abundance of secondary BAs. Although not significant, secondary BA levels were higher in those with no recurrence receiving FMSL-brpk compared to those with documented recurrence within 8 weeks (*p* = 0.08). Notably, factors associated with non-recurrence were the early engraftment of FMSL-brpk (*p* < 0.05) and increased secondary BAs (*p* < 0.0001) [71]. ECOSPOR provided a strong mechanistic basis for the administration of live, purified firmicutes through the demonstration that early engraftment with FMSL-brpk was associated with reduced rCDI rates and a minimal AE profile.

Two phase 3 trials assessed the therapeutic efficacy of FMSL-brpk: ECOSPOR III and ECOSPOR IV. ECOSPOR III included 182 patients who had three or more episodes of CDI within 12 months. The primary efficacy endpoint was CDI recurrence up to 8 weeks after treatment initiation. FMSL-brpk was found to be superior to the placebo in reducing rCDI: 12% vs. 40%, respectively; difference 28%; RR 0.31 95% CI 0.18–0.58; *p* < 0.001) [69]. Similar results were observed irrespective of the initial antibiotic used to treat CDI. While not statistically significant, FMSL-brpk led to less frequent rCDI when stratified by age: age < 65 years: RR 0.24, 95% CI 0. 07–0.78 and age ≥ 65 years: RR 0.36; 95% CI 0.18–0.72 [69]. AEs related or possibly related to the study drug or placebo occurred in slightly more than half of the patients in each group, the majority of which were mild to moderate GI disorders (e.g., flatulence, abdominal pain, constipation, diarrhea) [69].

In a secondary analysis of ECOSPOR III with an extended follow-up through 24 weeks, the rate of rCDI nearly doubled compared to the 8-week follow-up results but was still significantly improved in the FMSL-brpk group (21.3% FMSL-brpk vs. 47.3% placebo, RR 0.46; 95% CI 0.30–0.73; *p* < 0.001) [72]. Overall, FMSL-brpk demonstrated durable efficacy with reduced rCDI rates and was well tolerated through 24 weeks.

In a post hoc analysis, rates of rCDI through week 8 were analyzed for the following subgroups: Charlson comorbidity index score categories (0, 1–2, 3–4, ≥5); baseline creatinine clearance (>30, 30–50, >50 to 80, > 80 mL/min); the number of CDI episodes (three and ≥four); exposure to non-CDI targeted antibiotics; and acid-suppressing medications at baseline. Across all subgroups, FMSL-brpk was associated with a lower relative risk of CDI recurrence compared to the placebo, irrespective of baseline characteristics [73]. The results from this post hoc analysis illustrate the potential benefit of FMSL-brpk for complex and at-risk patients for rCDI.

ECOSPOR IV was an open-label, single-arm, phase 3 trial conducted in two cohorts. Cohort 1 included patients from the ECOSPOR III trial who had experienced CDI recurrence within 8 weeks after treatment with FMSL-brpk or a placebo. Cohort 2 were de novo patients with at least one CDI recurrence. The primary endpoint was safety tolerability up to 24 weeks after dosing. The secondary endpoint was CDI recurrence up to 4, 8, 12, and 24 weeks after dosing. The overall incidence of treatment-emergent AEs was 54%, but similar to previous trials, most were mild to moderate and gastrointestinal: diarrhea (22.8%), flatulence (7.6%), abdominal pain (6.8%), urinary tract infection (4.9%) and fatigue (4.6)% [74]. Notably, none of the urinary tract infections were caused by species included in FMSL-brpk. With respect to efficacy, 8.7% of patients in cohort 1 and 8.1% in cohort 2 had recurrent CDI. The rate of CDI recurrence remained low throughout the 24 weeks, achieving a sustained clinical response rate of 86.3% (95% CI 81.6–90.2%) [74]. An analysis based on the selected baseline characteristics demonstrated a low rate of CDI recurrence irrespective of age, CDI antibiotic treatment, sex, or the number of prior CDI episodes. ECOSPOR IV confirmed the durability of response and safety of FMSL-brpk through 24 weeks.

The FDA approval of FMSL-brpk was largely based on the results of ECOSPOR III, which demonstrated a significant reduction in rCDI in patients at increased risk for CDI recurrence and hospital admission, which included patients aged ≥65 years, those who were immunocompromised, and those who had malignancies or GI disorders. While ECOSPOR IV was primarily designed to assess tolerability, the results of this trial further supported the approval of FMSL-brpk through its demonstration of durable efficacy and minimal AE profile through 24 weeks. Compared to traditional FMT where routes of delivery vary from a nasogastric tube, nasojejunal tube and colonoscopy to a retention enema, FMSL-brpk offers a non-invasive, convenient route of oral administration, which may be a more comfortable option for microbiota-based therapies to some patients.

#### 3.4.5. VE303

VE303 is a defined consortium product composed of eight nonpathogenic, nontoxigenic, and commensal strains of Clostridia selected for their ability to provide colonization resistance to *C. difficile*. Under the current good manufacturing processes, it is produced from pure, clonal bacterial cell banks to create a standardized drug product in powdered form intended for oral administration. Unlike FMT, the manufacturing process of VE303 bypasses the need for it to be sourced directly from donor fecal material of inconsistent composition. VE303 is not currently an FDA-approved product, but in May 2023, the US FDA granted Fast Track designation to VE303 for the prevention of rCDI.

VE303 first demonstrated promise in VE303-002, a double-blind placebo-controlled trial. The study included patients with one or more prior CDI episodes within 6 months of randomization and included patients with primary CDI at high risk for recurrence defined as aged ≥75 years or aged ≥65 years, with at least one of the following prespecified risk factors for recurrence: kidney dysfunction, the regular use of a proton pump inhibitor, or a history of CDI > 6 months previously [75]. This trial was a dose-finding study in which study participants were divided into three groups: high-dose VE303 (8.0 × 10^9^ CFs), low dose (1.6 × 10^9^ CFUs), or placebo. VE303 was administered within 24 h after completing antibiotic treatment. The most robust difference in CDI recurrence was in the high-dose VE303 group compared to the placebo, 13.8% (4/29) vs. 45.5% (10/22), respectively (ARR 30.5%, 90% CI 11% to 52%) [75]. Most patients experienced a sustained cure through 24 weeks with only two CDI recurrences reported, suggesting a durable effect. No significant difference was found between low-dose VE303 and the placebo. All recurrences occurred by day 11 in the high-dose group. VE303 was generally well tolerated, with most treatment-related AEs being mild in intensity and primarily gastrointestinal (e.g., diarrhea, abdominal pain, flatulence, and vomiting). Importantly, no bacterial infection or AEs of interest were noted in this study. The authors suggested that the absolute risk reduction of 30.5% in the high-dose VE303 group is more favorable comparably to other CDI therapeutic options including FMT (28%), bezlotoxumab (10%), and FM, jslm (12.3%) [75]. The results of VE303-002 are promising and provide the rationale for pursuing high-dose VE303 002 in the larger-scale phase 3 trial, which is still undergoing recruitment.

The phase 3 trial of VE303, RESORATIiVE303, is designed to assess the safety and recurrence rate of CDI at week 8 among study participants who undergo a 14-day treatment with either VE303 or a placebo. The results of this trial could lead to changes in the management of CDI by providing a new oral option for CDI.

### 3.5. Other Preventative Strategies

Over the years, multiple strategies have been deployed to prevent the emergence of *C. difficile*. The scope of these strategies vary from impact at the individual level to impact at the population level. Population-based strategies include establishing antimicrobial stewardship programs and infection prevention programs. The latter relies on non-pharmaceutical-based strategies and aims at curtailing the spread of CDI between infected and healthy individuals by isolating infected individuals through established procedures in healthcare settings. In this section, we will discuss pharmaceutical-based strategies to prevent *C. difficile* infection.

#### 3.5.1. Antimicrobial Stewardship

Antimicrobial stewardship programs (ASPs) aim at ensuring the judicious use of antimicrobials across healthcare settings. These programs formulate policies and procedures to optimize the use of antimicrobials. The benefits of ASPs include improving the local microbial ecology and preventing emergence of multi-drug resistant organisms and *C. difficile* infection (CDI) at the individual level. The judicious use of antimicrobials results in the retention of microbial diversity and an increase in colonization resistance against *C. diffcile*. At our own institution, we have seen an impressive reduction in healthcare-acquired CDI as our appropriate antimicrobial usage has improved over the years. Over a period of 7 years, i.e., from 2016 through 2023, we saw a 44.1% (1.36 to 0.76) reduction in the CDI standard infection ratio (SIR) and a 44.3% (69 cases to 44 cases) reduction in the absolute number of CDI cases. At the same time, we saw a 9% improvement in the appropriate usage of our broad-spectrum antimicrobials.

Antimicrobial use is pervasive in healthcare settings. In the United States, approximately 50% of patients in hospitals receive antimicrobials [76]. The appropriate usage of antimicrobials is lifesaving in conditions like sepsis, which impacts 1.7 million adults in the United States annually [76]. At the same time, the incidence of CDI is high at 116 cases per 100,000 persons, and 56% of CDI cases have received antimicrobials in the prior 12 weeks [77]. Therefore, as clinicians, it is helpful to know the risk associated with each class of antibiotics, so we can weigh the risk and benefits in clinical situations to harness the maximum benefits of antimicrobials. Intuitively, broader-spectrum antibiotics will be associated with a greater risk of CDI compared with narrower-spectrum antibiotics. This is supported by a large cohort study in an inpatient setting, where carbapenems had the highest risk, followed closely by piperacillin–tazobactam and cefepime. In the same study, the lowest risk was noted for doxycycline and daptomycin [78]. Similarly, in the outpatient setting, the lowest risk of CDI was with doxycycline, minocycline and tetracycline, while the highest risk was with clindamycin, followed by cephalosporins and fluroquinolones [79]. With such granular information, clinicians can ensure the responsible use of antimicrobials and prevent CDI.

#### 3.5.2. Probiotics

Probiotics have been studied for their primary prevention and secondary prevention of CDI. The results across various randomized controlled trials are conflicting.

A multicenter randomized, double-blind, placebo-controlled, pragmatic, efficacy trial was conducted to evaluate the efficacy of microbial preparations, i.e., lactobacilli and bifidobacteria in hospitalized individuals who were 65 years of age or older. The investigators did not find evidence that the microbial preparations of lactobacillus and *Bifidobacterium* were effective in preventing antibiotic-associated diarrhea (AAD) or *C. difficile* diarrhea (CDD). The primary outcomes in this study were the occurrence of AAD within 8 weeks and CDD within 12 weeks of enrollment. AAD occurred in 10.8% (159/1470) of participants who received the probiotic and in 10.4% (153/1471) of the participants in the placebo group. However, CDD was observed in 0.8% (12/1470) of the participants in the microbial preparation group and in 1.2% (17/1471) of the participants in the placebo group. For both primary outcomes, i.e., AAD and CDD, the difference between the microbial preparation arm and the placebo arm was not statistically significant [80].

Similarly, another multicenter, double-blind, placebo-controlled, randomized trial was conducted to assess the impact of a probiotic, *Lactobacillus casei* DN114001, on AAD and CDD in individuals more than 55 years of age. The investigators concluded that there was no beneficial effect of this formulation of probiotic on AAD, i.e., 19.3% (106/549) developed AAD in the probiotic group and 17.9% (103/577) of the participants in the placebo group. The impact of this intervention on CDD was challenging to quantify since there had been multiple changes in antimicrobial stewardship practices and modifications in nursing practices and infection control practices across healthcare facilities [81].

Conversely, in a retrospective analysis by Dudzicz et al., prophylaxis with the probiotic *Lactobacillus plantarum* 299V (LP299v) in high-risk groups can prevent primary CDI [82].

A Cochrane meta-analysis of 31 RCTs showed that probiotics are effective in preventing primary CDI in patients receiving antimicrobials, with the number needed to benefit at 42, with moderate certainty. A post hoc subgroup analysis conducted to explore the heterogeneity of the trials showed a benefit with moderate certainty only when the CDI baseline risk was more than 5% (NNTB = 12). In US hospitals, typically, the risk of CDI is less than 5% [83,84]. On the contrary, a multicenter study did not show a beneficial impact of probiotics on the primary prevention of CDI [85]. The most commonly used probiotics in these meta-analyses included *Saccharomyces boulardii*, *Lactobacillus rhamnosus* GG, *Lactobacillus casei* DN11400 and two types of probiotic mixtures containing *Lactobacillus acidophilus* and *Bifidobacterium*.

The role of probiotics in secondary prevention is even more controversial. In a meta-analysis, the use of probiotics for secondary prevention did not reach statistical significance to make a firm conclusion [86].

In conclusion, currently available probiotics have not proven to be effective in preventing CDI. This is likely due to the heterogeneity in studies regarding setting (inpatient versus outpatient), probiotic strains, and optimal dose, along with our limited understanding of the mechanisms by which probiotics exert their action [84].

#### 3.5.3. *C. difficile* Vaccine

Even though *C. difficile* has been designated as one of the five urgent threats by the CDC, there is currently no approved vaccine for the prevention of primary CDI or recurrent CDI.

A recent phase 3, randomized, placebo-controlled trial studying the efficacy of a three-dose series (0, 1 and 6 months) of PF-06425090 in primary CDI prevention was completed. The candidate vaccine PF-06425090 is a genetically detoxified toxin *C. difficile* vaccine formulated with modified toxin A and toxin B. This study, also known as the CLOVER study, CLOstridium difficile Vaccine Efficacy tRial, included participants if they were 50 years of age or older and were considered at high risk for CDI. Criteria for high risk included individuals who were in a nursing home or skilled nursing facility, had healthcare exposure in the last year and had received antibiotics in the past 12 weeks. This was a large study with more than 7000 participants in each of the arms, vaccine and placebo. The primary endpoint was the first episode of CDI 14 days or more after dose 3. Following the third dose, 17/7724 developed CDI in the vaccine arm and 25/7818 developed CDI in the placebo arm resulting in a vaccine efficacy of 31% (−38.7 to 66.6). The primary efficacy endpoint for a vaccine efficacy of more than 20% was not met. Adverse events were similar in both groups, ranging from mild to moderate [87].

Several other vaccine candidates are in development including a toxoid-based vaccine, which is currently in a phase 3 clinical trial. In this study, healthy adults who are within the age range of 65 years to 85 years are included. Based on the results thus far, a regimen of three doses (0, 1, or 6 months) is well tolerated and induces a robust neutralizing antibody response [88]. Another potential candidate is a protein-based vaccine candidate, VLA 84, which is also in the pipeline, though a phase 3 clinical trial has been put on hold for now [89]. In addition, more recently, a messenger RNA (mRNA)–lipid nanoparticle vaccine targeting *C. difficile* toxins and virulence factors was developed in a mouse model. This vaccine provided protection against both primary and recurrent infection in mouse models [90]. Of note, this study was published in October 2024, which is outside the time frame of our literature search. However, given the relative importance of this study, we feel it is imperative to mention the study in this review.

Based on prior vaccine work, toxoid-based vaccines will not prevent transmission. Hence, consideration has been given to non-toxoid-based vaccine candidates, such as VLA 84 [89]. At this time, the evidence for vaccine use for *C. difficile* prevention is still in the investigational phase.

#### 3.5.4. Antibiotic Prophylaxis

As the risk of rCDI significantly increases with each subsequent CDI episode, it is important to identify risk factors strongly associated with recurrence and mitigate the risk where feasible. Risk factors most associated with CDI include advanced age, an immunocompromised status, inflammatory bowel disease, a history of a prior CDI episode, and the administration of systemic antibiotics [22]. While most risks cannot be modified, antibiotic use can be minimized to help prevent the recurrence of CDI. However, some antibiotic courses are unavoidable and necessary; therefore, CDI prophylaxis has been pursued as the logical approach to reduce the risk of recurrence in this setting.

The IDSA/SHEA guidelines cite that there is insufficient evidence to either extend the duration or restart anti-CDI treatment in the setting of systemic antibiotic therapy for non-CDI indications [22]. Alternatively, the ACG guidelines make conditional recommendations to consider long-term suppressive oral vancomycin in patients with rCDI who are ineligible to receive FMT, who have relapsed after FMT, or are requiring ongoing courses of antibiotics. The ACG also recommends considering oral vancomycin prophylaxis during systemic antibiotic use for patients at a high risk of recurrence [33].

The data regarding *C. difficile* prophylaxis are largely retrospective and observational in design and therefore at a high risk for bias [91]. Additionally, these studies have heterogenous methodologies, including their patient population; indications for use such as primary versus secondary prevention; prophylaxis regimens including antibiotic, dose, and duration; and follow-up requirements. The current data lack a standardized approach to prophylaxis, which limits the clinical utility of their conclusions. Furthermore, the use of antibiotics can disrupt the gut microbiome, leading to a loss of diversity predisposing patients to colonization and infection with *C. difficile* up to 90 days from antibiotic discontinuation. As a result, retrospective, observational studies may be inherently plagued with attrition bias due to shorter follow-up periods that underestimate the rate of CDI [91].

With respect to *C. difficile* prophylaxis, only two RCTs evaluating vancomycin and fidaxomicin have been published, and each one evaluated the role of antibiotics as the primary prophylaxis. A more recent randomized, prospective, open-label study compared the efficacy of prophylaxis with vancomycin 125 mg once daily during the course of systemic antibiotics continued for 5 days after discontinuation versus no prophylaxis. Patients were considered for prophylaxis if they met one of the following high-risk criteria and were receiving systemic antibiotics at the time: aged ≥60 years or hospitalized ≤30 days prior to index hospitalization and received antibiotics during that prior hospitalization. No CDI events (0/50) occurred in the oral vancomycin prophylaxis group compared to 12% (6/50) in the no-prophylaxis group, which was evaluated up to 3 months post discharge (*p* = 0.03) [92]. Given concerns with oral vancomycin’s effect on the gut microbiome and the selection and overgrowth of vancomycin-resistant enterococci (VRE), the authors of this study also evaluated new VRE colonization defined as the isolation of VRE by a perirectal swab prior to hospital discharge. No patients developed new VRE colonization. Notably, evidence for the selection of VRE following the use of vancomycin treatment for CDI is conflicting. There are prospective studies and time series analyses demonstrating the emergence of VRE following the use of oral vancomycin. Conversely, there are studies showing a lack of emergence of VRE in similar settings, thus highlighting the conflicting body of evidence [93,94]. Moreover, a double-blind RCT of fidaxomicin 200 mg daily for the prophylaxis of CDI in hematopoietic stem cell transplants receiving fluroquinolone prophylaxis confirmed that CDI was significantly lower in the fidaxomicin recipients (4.3%) than in the placebo recipients (10.7%) during the 60-day follow-up [*p* = 0.0014] [95].

There are limited data to universally recommend antibiotic prophylaxis for either primary or secondary prevention. In the absence of long-term data, antibiotic prophylaxis may be considered on a case-by-case basis if its benefit outweighs the risk. Such cases may include elderly and immunocompromised patients with a prior history of CDI who are unable to receive microbiota-based therapies. Based on the available RCTs, both vancomycin and fidaxomicin are promising prophylactic strategies that have reduced CDI in high-risk patients. While more evidence exists for vancomycin, fidaxomicin is a narrower-spectrum agent with less disruption to the gut microbiome and is associated with a lower rate of recurrence. Additionally, previous studies have demonstrated that prolonged vancomycin use may be associated with complications such as subsequent *Candida* spp. and enteric bacterial bloodstream infections [96]. Therefore, fidaxomicin may be the preferred option to minimize alterations to intestinal microbiota [96]. While optimal prophylactic dosing is still to be elucidated, generally, one aims to use the smallest effective dose. Based on RCTs, the most reasonable dosing options are the following: oral vancomycin 125 mg daily or fidaxomicin 200 mg daily for the duration of the systemic antibiotic course.

## 4. Conclusions and Future Directions

Since its discovery in 1935, *C. difficile* has become a leading cause of healthcare-associated infections in the US, increasing the morbidity and mortality of patients in healthcare settings. Antibiotic exposure is one such modifiable risk factor for CDI. Consequently, at the population level, the effective implementation of an antimicrobial stewardship program can be useful in preventing *C. difficile* infection. Stewardship interventions limit the use of unnecessary antimicrobials and minimize the frequency, duration and number of antimicrobials prescribed to help reduce the risk of CDI.

Once CDI is diagnosed, it is imperative to distinguish disease severity and the number of prior CDI occurrences as this can determine how it is treated (See Figure 2). In addition to standard-of-care antibiotics, monoclonal antibodies like bezolotoxumab have been developed to reduce rCDI and can be considered in patients with risk factors for recurrence such as those who are ≥65 years, are immunocompromised, have severe CDI or had a prior CDI in the past 6 months. Patients with two or more episodes of CDI are at a higher risk of recurrence, with rCDI occurring upwards of 65%. Management can be challenging in this setting as standard-of-care antibiotics for CDI can further disrupt the gut microbiome. Therefore, microbiota restoration therapies should be explored. While FMT is the only guideline-directed option, newer live biotherapeutic options have been FDA-approved since the guidelines were last updated. Fecal microbiota, live-jslm and fecal microbiota spores, live-brpk offer a safe, standardized, and more convenient approach to microbiota-based therapies compared to FMT. Real-world experience with these new agents will help determine their place in therapy.

To have a true impact on the huge morbidity associated with CDI and rCDI, our emphasis needs to shift from treatment strategies to prevention strategies. In recent years, our understanding of the progression from asymptomatic CDI to symptomatic CDI has improved with the advent of fecal metabolomics. For instance, we have learnt that microbiomes of asymptomatic CDI are richer in species of the Clostridia class relative to symptomatic CDI microbiomes. In addition, the microbiome of asymptomatic CDI is enriched with carbohydrate compounds compared to symptomatic CDI microbiomes. As our understanding of the pathogenesis of CDI evolves, we should aim to identify microbial signatures in the gut that represent an unhealthy microbiome and can predispose one to CDI. If we can identify such microbial signatures early, especially in our vulnerable populations, we could leverage preventive strategies like *C. difficile* vaccine and probiotics to prevent the emergence of CDI. At this time, for both these preventive strategies, the evidence is either not definitive or it is in the investigational phase. However, as our understanding of the factors associated with gut dysbiosis matures, these preventive strategies (along with others) can be leveraged to reduce the *C. difficile*-associated morbidity and ultimately eliminate *C. difficile* from the CDC urgent threat list.

## Figures and Tables

**Figure 1 antibiotics-14-00017-f001:**
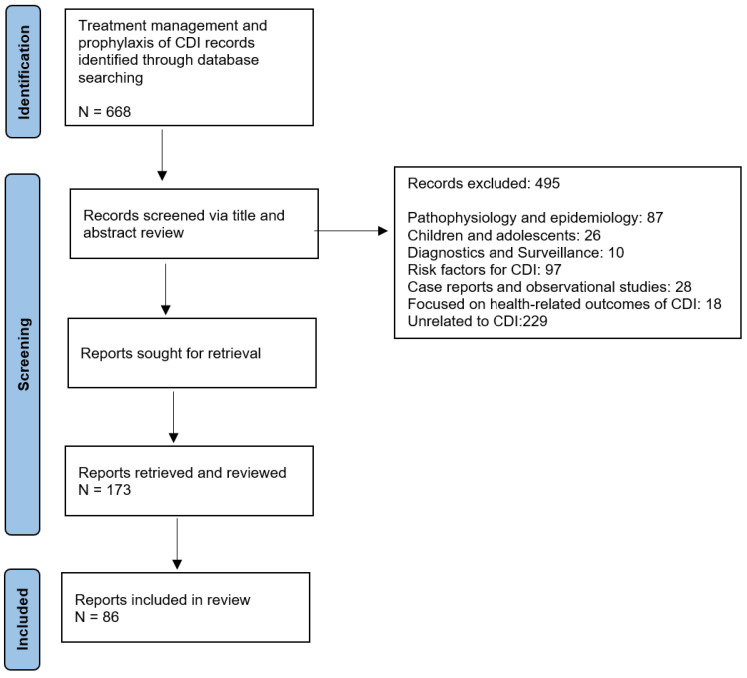
Literature review flow diagram.

**Figure 2 antibiotics-14-00017-f002:**
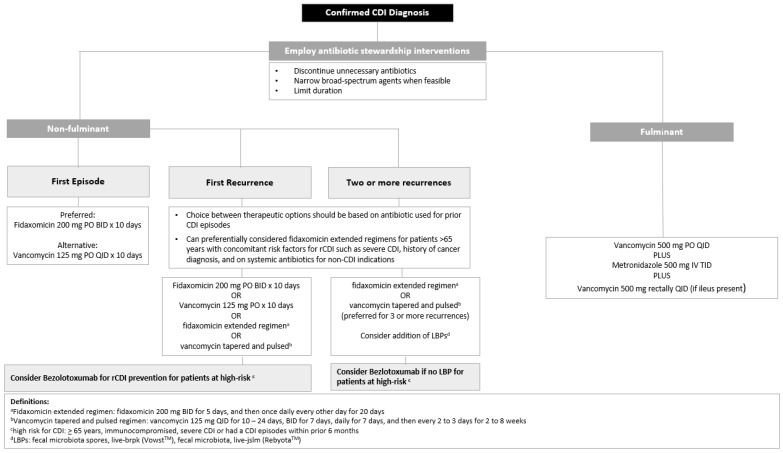
Treatment options for CDI based on severity and number of prior CDI episodes.

**Table 1 antibiotics-14-00017-t001:** **The** 2017 and 2021 IDSA/SHEA and 2021 ACG treatment recommendations for adult patients with CDI.

	IDSA/SHEA	ACG
**Initial CDI episode**	Fidaxomicin 200 mg BID for 10 days (preferred) - OR Vancomycin 125 mg QID for 10 days - OR Metronidazole 500 mg TID for 10–14 days (non-severe CDI only and if vancomycin or fidaxomicin is unavailable) Adjunctive: bezlotoxumab 10 mg/kg IV once ^a^	**Non-Severe** Vancomycin 125 mg QID for 10 days - OR Fidaxomicin 200 mg BID for 10 days - OR Metronidazole 500 mg TID for 10 days (low-risk patients ^c^) **Severe** Vancomycin 125 mg QID for 10 days - OR Fidaxomicin 200 mg BID for 10 days - OR FMT ^d^ Adjunctive: bezlotoxumab 10 mg/kg IV once ^b^
**First recurrence**	Fidaxomicin 200 mg BID for 10 days (preferred) - OR Fidaxomicin 200 mg BID for 5 days and then once daily every other day for 20 days (preferred) - OR Vancomycin tapered and pulsed (e.g., 125 mg QID for 10–14 days, BID for 7 days, daily for 7 days, and then every 2 to 3 days for 2 to 8 weeks) - OR Vancomycin 125 mg QID for 10 days (if metronidazole was used for primary infection) Adjunctive: bezlotoxumab 10 mg/kg IV once ^a^	Vancomycin tapered and pulsed - OR Fidaxomicin 200 mg BID for 10 days (unless fidaxomicin was used for primary infection) Adjunctive: bezlotoxumab 10 mg/kg IV once ^b^
**Second or subsequent recurrence**	Fidaxomicin 200 mg BID for 10 days - OR Fidaxomicin 200 mg BID for 5 days and then once daily every other day for 20 days - OR Vancomycin tapered and pulsed (e.g., 125 mg QID for 10–14 days, BID for 7 days, daily for 7 days, and then every 2 to 3 days for 2 to 8 weeks) - OR Vancomycin 125 mg QID for 10 days followed by rifaximin 400 mg TID for 20 days - OR FMT Adjunctive: bezlotoxumab 10 mg/kg IV once ^a^	FMT Adjunctive: bezlotoxumab 10 mg/kg IV once
**Fulminant CDI**	Vancomycin 500 QID - PLUS Metronidazole IV 500 mg TID - PLUS Vancomycin rectal 500 mg in 100 mL 0.9% sodium chloride QID (if ileus present)	Vancomycin 500 QID for the first 48–72 h, followed by 125 mg QID - PLUS Metronidazole IV 500 mg TID - PLUS Vancomycin rectal 500 mg in 100 mL 0.9% sodium chloride QID (if ileus present) - OR FMT ^d^

^a^ IDSA/SHEA recommends considering the addition of bezlotoxumab in patients who are age > 65 years, are immunocompromised, have severe CDI or had CDI episodes within prior 6 months. ^b^ ACG recommends considering bezlotoxumab in patients who are age > 65 years and have one of the following risk factors for recurrence: CDI episode within prior 6 months, immunocompromised or severe CDI. ^c^ Low-risk patients defined as younger outpatients with minimal comorbidities. ^d^ ACG recommends considering FMT for patients with CDI refractory to antibiotic therapy, particularly when deemed to be poor surgical candidate.

**Table 2 antibiotics-14-00017-t002:** FDA-approved microbiota restoration therapies.

	Fecal Microbiota, Live-Jslm (Rebyota^TM^)	Fecal Microbiota Spores, Live-Brpk (Vowst^TM^)
Route	Rectal enema	Oral capsule
Dose	Single dose of 150 mL rectally	4 capsules orally once daily for 3 days
Antibiotic wash-out period	Administered 24 to 72 h after CDI antibiotic therapy discontinuation	First dose administered 24 to 72 h after CDI antibiotic therapy discontinuation
Administration	Thaw by placing product in refrigerator (2–8 °C) for 24 h Patient should empty their bladder and bowel, if possible, prior to administration Keep the patient in the left-side positive or knee–chest position for up to 15 min after administration to minimize cramping and expulsion	Prior to the first dose, patient should drink 296 mL (10 oz) of magnesium citrate (or polyethylene glycol for patients with renal dysfunction) on the day before and at least 8 h prior to taking the first dose Should be administered before the first meal on an empty stomach
Storage	Ultracold freezer (−60 to −90 °C) - OR- Refrigerator (2 to 8 °C) for up to 5 d (including thaw time) Do not freeze after thawing	Original packaging at 2 to 25° C Do not freeze

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
