# Peer review of "A Review of Therapies for *Clostridioides difficile* Infection"

_antibiotics, 2024, doi:10.3390/antibiotics14010017_

Round 1
Reviewer 1 Report
Comments and Suggestions for Authors
Dear Authors,
I sincerely thank you for the opportunity to review this manuscript, that I read with great interest. I personally believe that this manuscript is precisely written and structured, with a solid methodology. I just have few suggestions in order to further improve its quality.
- Line 9: there is a little typo within the word "States";
- Lines 413-414: I suggest to better clarify the distinction between EU countries and UK, since the sentence as it currently is could be confusing.
- Line 735: could you add a citation about the sentence "approximately 50% of patients in hospitals receive antimicrobials"? Does it refer to the United States?
- Line 750: is this paragraph related specifically to particular probiotics, or only focused on probiotics in general?- Line 750ve
-jslm fecal microbiota suspension
I express my congratulations for this work, which I already consider a valid resource that should be used for studying this topic.
In conclusion, I believe that this manuscript can be considered for publication in Antibiotics after minor revisions. I remain at your disposal.
Best Regards
Reviewer 2 Report
Comments and Suggestions for Authors
The review article entitled “A Review of Therapies for Clostridioides difficile Infection” in very interesting to read. I recommend this manuscript for publication after minor revision.
Revision points
Introduction is not sufficient and it does not reflect the whole manuscript. Introduction must be hypothesis driven.
Please use nomenclature carefully, Line 20: correct bacillus as, Bacillus
What is the status of this disease, “Clostridioides difficile infection (CDI)”, other than US?
This disease was only reported in US?
What is the major reason for “Recurrent Clostridioides difficile infection” in humans?
It is desirable to include recent advances on Clostridioides difficile treatment.
Line 726: Please use italics
Line 750: Additional information required. Deep discussion required, please refer the following review
https://doi.org/10.3390/nu16050671
765: Italics?
Please include the following
Recent development on vaccine development. Will PHILADELPHIA is useful? Please discuss about mRNA vaccines?
Reviewer 3 Report
Comments and Suggestions for Authors
Thank you for the invitation to review the submission by Faiza Morado and Neha Nanda (Manuscript ID: antibiotics-3364015). The authors searched the PubMed database, summarized CDI-related papers within the past five years, and gave a clinical recommendation about selecting and using commercial medications against C. difficile. Along with all the clinical medications, the authors also summed up and shared thoughts on preventive strategies. This review is generally well organized and has provided up-to-date clinical therapies against CDI. From my point of view, this review could serve as a guideline or at least reference protocol for clinicians to refer to. Therefore, I would recommend acceptance after addressing several minor concerns of mine.
1. I would recommend the authors include a prefix at the very beginning of their review, so that the readers can locate the contents of their interest more intuitively and quickly.
2. Page 3, please reorganize the 4th paragraph under the “Metronidazole” subtitle, maybe integrate it with the “Vancomycin and Fidaxomicin” section, as it mainly focuses on the advantage of vancomycin over metronidazole.
3. The authors mentioned oral administration of vancomycin has risks in selecting vancomycin-resistant enterococci (VRE). I am very curious if there are clinical reports about the occurrence of VISA/VRSA during the administration of vancomycin to CDI patients. Could you discuss the occurrence of antimicrobial resistance in these pathogens more?
4. For the FMT/LBP treatment, I am wondering if the transplantation of the donor’s microbiota could affect recipients’ liver-gut axis or brain-gut axis, which could endanger their health in the long term. Could you look up more related research and share your thoughts on this subject?
5. As to the gut microbiome, there are a lot of ongoing multi-omics studies. Could the authors discuss more about the relation between gut microbiome and CDI? Is it possible to tell the composition of “healthy” microbiomes from “unhealthy” ones in the clinical settings as of today?
